# From Ocean to Medicine: Pharmaceutical Applications of Metabolites from Marine Bacteria

**DOI:** 10.3390/antibiotics9080455

**Published:** 2020-07-28

**Authors:** José Diogo Santos, Inês Vitorino, Fernando Reyes, Francisca Vicente, Olga Maria Lage

**Affiliations:** 1Department of Biology, Faculty of Sciences, University of Porto, Rua do Campo Alegre s/n, 4169-007 Porto, Portugal; ines.rjv@gmail.com (I.V.); olga.lage@fc.up.pt (O.M.L.); 2Interdisciplinary Centre of Marine and Environmental Research, University of Porto, Terminal de Cruzeiros do Porto de Leixões, Avenida General Norton de Matos, S/N, 4450-208 Matosinhos, Portugal; 3Fundación MEDINA, Centro de Excelencia en Investigación de Medicamentos Innovadores en Andalucía, Avenida del Conocimiento, 34 Parque Tecnológico de Ciencias de la Salud, 18016 Granada, Spain; fernando.reyes@medinaandalucia.es (F.R.); francisca.vicente@medinaandalucia.es (F.V.)

**Keywords:** marine natural products, antimicrobials, antivirals, anticancer, drug discovery, bioactive marine bacteria, antimicrobial resistance, cancer incidence, One Health

## Abstract

Oceans cover seventy percent of the planet’s surface and besides being an immense reservoir of biological life, they serve as vital sources for human sustenance, tourism, transport and commerce. Yet, it is estimated by the National Oceanic and Atmospheric Administration (NOAA) that eighty percent of the oceans remain unexplored. The untapped biological resources present in oceans may be fundamental in solving several of the world’s public health crises of the 21st century, which span from the rise of antibiotic resistance in bacteria, pathogenic fungi and parasites, to the rise of cancer incidence and viral infection outbreaks. In this review, health risks as well as how marine bacterial derived natural products may be tools to fight them will be discussed. Moreover, an overview will be made of the research pipeline of novel molecules, from identification of bioactive bacterial crude extracts to the isolation and chemical characterization of the molecules within the framework of the One Health approach. This review highlights information that has been published since 2014, showing the current relevance of marine bacteria for the discovery of novel natural products.

## 1. Health Challenges Faced Today

### 1.1. The Health and Economic Burden of Antibiotic Resistance

With the increase of global travel and the incorrect and excessive use of antibiotics, namely for animal production, a drastic rise in antibiotic resistance is being observed in bacterial populations throughout the world [1]. Infections with resistant organisms not only entail higher morbidity and mortality, but are also more expensive to treat and result in longer hospital stays, which places a greater burden on healthcare systems [2]. These facts make the rise in antibiotic resistance one of the greatest health challenges of the century. Based on World Health Organization (WHO) reports on the global rise of resistance in pathogenic bacteria, the Global Antimicrobial Resistance Surveillance System (GLASS) report [3,4] forecasts an alarming future for humankind. Patients with third-generation cephalosporin-resistant *Escherichia coli* infections have experienced a two-fold increase in mortality attributable to bacterial infections and a significant mortality increase is also observed in patients with methicillin-resistant *Staphylococcus aureus* infections. Facts like these point to the imperative need for the discovery of novel antibiotics to fight these super bacteria. Moreover, the global priority list of antibiotic-resistant bacteria to guide research, discovery, and development of new antibiotics also released by the WHO, specifies the top priorities for research and development of novel antibiotics. Critical bacteria comprise carbapenem-resistant Enterobacteriaceae (e.g., *Klebsiella pneumoniae*, *E. coli*, *Enterobacter* sp., *Serratia* sp., *Proteus* sp., *Providencia* sp. and *Morganella* sp.) and *Acinetobacter baumannii, Pseudomonas aeruginosa*, as well as 3rd generation cephalosporin-resistant Enterobacteriaceae. High priority bacteria comprise vancomycin-resistant *Enterococcus faecium*, methicillin-resistant and/or vancomycin-resistant *S. aureus*, clarithromycin-resistant *Helicobacter pylori*, fluoroquinolone-resistant *Campylobacter* sp. and fluoroquinolone-resistant *Salmonella* sp. and *Neisseria gonorrhoeae*, as well as 3rd generation cephalosporin-resistant *N. gonorrhoeae*. Medium priority bacteria include fluoroquinolone-resistant *Shigella* sp., ampicillin-resistant *Haemophilus influenzae* and penicillin-non-susceptible *Streptococcus pneumoniae* [5].

Regarding fungi, infections by resistant *Candida* spp. and *Aspergillus* spp. are increasing. The Centers for Disease Control and Prevention (CDC) recently published data showing that 7% of all bloodstream *Candida* isolates are resistant to fluconazole [6]. Moreover, the CDC also reports that about 90% of *Candida auris* isolates are resistant to fluconazole and 33% to amphotericin B [7]. *Aspergillus* infections, although only common in immunosuppressed individuals, are also becoming harder to fight, as azole resistance is also being observed [8,9,10].

In addition, the spread of antimicrobial resistance entails a vast economic loss. It is estimated that the economic impact of these diseases will reach up to US$3.5 billion in losses per year due to deaths caused by resistant microorganisms in the next 30 years just in Europe, North America and Australia [11].

Likewise, there is a significant rise of resistance in protozoal parasites, belonging to the genera *Plasmodium*, *Trypanosoma* and *Leishmania* [12,13,14]. These parasites are responsible for malaria, African and American trypanosomiasis and leishmaniasis, respectively, which disproportionally affect the poorest and least prepared nations. This is aggravated by the fact that few drugs are effective to combat these diseases. For example, in the 1960s, *Plasmodium* developed resistance to chloroquine, a first line treatment drug against malaria [15] and only a few families of drugs remain capable of treating malaria. However, in Southeast Asia, the infectious agents of malaria already show signs of resistance to artemisinin [16], which was discovered in 1975 and is currently the main treatment for malaria [17].

### 1.2. Impacts of Viral Infectious Outbreaks

Viruses are infectious agents that can only reproduce inside living cells. In their most basic form, they consist of only genetic material, such as DNA or RNA (retroviruses) and a protein coat, the capsid. This basic plan can then be supplemented with other components, like an outer envelope of lipid coating or, for retroviruses, the reverse transcriptase enzyme [18].

Viruses have a great capacity to spread causing viral infections in humans. Due to horizontal transmission, the most common mechanism of virus spreading in populations, we have witnessed several viral pandemics that have led to health and economic crises. Since December 2019, the world is in the grips of another pandemic outbreak, caused by the SARS-CoV-2 virus, which has already caused thousands of deaths and disrupted the global economy [19]. However, this is just the most recent of several pandemic and epidemic outbreaks caused by viruses, like those of Zika in 2015 [20], Ebola which flared up in West Africa in 2013 [21], the 2009 influenza pandemic caused by H1N1/09 and the severe acute respiratory syndrome (SARS), caused by SARS-CoV-1 in 2002. This last mentioned epidemic, like SARS-CoV-2, affected China, causing 774 deaths and 100 billion dollars in losses [22]. Moreover, humanity is still struggling with past pandemics, such as the HIV/AIDS pandemic, which still infects 37 million people around the world [23]. Furthermore, as novel viral diseases keep appearing, and antivirals are usually specifically targeted, research of new therapeutics is continuously needed [24].

### 1.3. “One Health” Framework

Many of these novel microbial threats are zoonotic in nature [25]. Zoonotic diseases are found to be spreading due to pressures on the environment like, for example, encroachment of human populations on animal territory. Moreover, animals are known reservoirs of endemic diseases also [26,27]. The “One Health” framework thus looks for combat diseases by a three-pronged strategy of communication, coordination and collaboration between human, animal, environmental health, and other relevant experts [28,29]. This integrated, interdisciplinary work is vital to understand and tackle emergent and re-emergent threats of infectious diseases, as it looks for environmental clues for discerning infectious threats and generating plans for confronting these threats.

### 1.4. Cancer Incidence and Mortality

As one of the most important public health problems, cancer is currently the second most frequent cause of death worldwide, being responsible for an estimated 9.6 million deaths in 2018 [30]. Between 2005 and 2015, it was the cause of one-third of all deaths worldwide [31]. For 2018, the GLOBOCAN database detailed that the major cancer types are of the lung, female breast and colorectum, which account for one-third of the cancer incidence and mortality [32].

Cancer is the generic name for a collection of more than 100 diseases due to uncontrolled division of cells that have developed over time. In cancer, the tightly controlled processes of cell division are deregulated by either the activation of proto-oncogenes or the inactivation of tumour suppressor genes. However, in order for a cell to develop into a cancerous tumour, other processes, such as apoptosis and senescence, cannot occur in the tumour cells. Cells must also have access to oxygen and nutrients [33]. Genetic factors can play an important part in the appearance of cancerous formations. This is the case of breast cancer and the tumour suppressing gene, *BRCA1*, as carriers are predisposed to life-time risks of up to 80% [34]. Furthermore, environmental factors, such as tobacco smoking, increase in urban pollution, increased intake of high caloric foods and longer lifespans all contribute to increasing the risk of cancer diagnosis [35]. Cancer is treated through the management of the risk factors and the use of antitumoral chemotherapeutic agents, surgery, radiotherapy or immunotherapy.

Antitumoral chemotherapeutics of microbial origin, many of which are produced by bacteria of the genus *Streptomyces*, are widely used in clinical practice and comprise mostly cytotoxic compounds that include the anthracycline daunomycin and its derivative doxorubicin, the polyketides aclarubicin and mithramycin, the glycopeptidic bleomycins A2 and B2, the chromopeptide actinomycin D and mitomycin C [36]. Nevertheless, the discovery of novel anticancer drugs remains a highly challenging endeavour. Not only must molecules have a specific effect, usually inhibitory which leads to cytotoxic effect on a target (or multiple targets), but toxic side effects need to be manageable [37]. This implies that many molecules must be extensively screened for side effects before their possible application. Natural products can reliably increase the diversity of bioactive molecular structures and, thus, increase the likelihood of developing novel therapeutics.

After pointing out the fundamental health risks faced by humankind, this review explores the contribution of marine bacterial derived natural products in the fight against cancer, antimicrobial resistance and viruses, with an update highlighting information from the literature published since 2014. Moreover, the technicalities needed for discoveries will be detailed.

## 2. Natural Products

### 2.1. The History of Marine Natural Products

Nature has long been the most important source of therapeutics. The use of poultices and mixtures of plant material to treat infections goes back to the early bronze age civilizations [38]. Building on the prior knowledge acquired, early medicine, pharmacology and chemistry started to develop therapeutics by studying nature. For example, extracts of willow bark (genus *Salix*), containing salicylic acid, which was identified to be the bioactive molecule present in the bark in 1828, were already used by Sumerians and Egyptians to treat inflammation and pain. In 1852 acetylsalicylic acid was first synthesised and in 1899, Bayer patented it as aspirin [39]. Ever since then, many terrestrial organisms and in particular plants have been sought for use as natural products. It was only around the 1970s, that attention was first given to the ocean as a source of useful natural products. As the oceans cover most of the Earth’s surface, they are home to a substantial portion of the world’s biodiversity [40] which lives in distinctive and varied conditions and has evolved through a long period of metabolic adaptations. When exploration of the ocean’s biodiversity and metabolic richness began, it resulted in the discovery of thousands of structurally unique bioactive marine natural products [41].

Initially, the exploitation of marine wildlife for natural bioactive products focused on a small number of organisms which included sponges, molluscs, tunicates and macroalgae [42]. These were shown to produce a very diverse range of unique molecular structures, like halogenated terpenes, polyketides and prostaglandins [43,44,45,46]. This diversity of bioactive structures is considered to be part of the defence, survival and predatorial strategies employed by these organisms, such as, for example, sponges, which are sessile, soft-bodied organisms, lacking morphological defences like biological armature or spines [47]. Thus, these organisms appeared to be a great source for the discovery of novel bioactive molecules. However, the bioactive molecules produced by these organisms can be present in quite small amounts. For example, halichondrin B (**1**) (Figure 1), a macrolide first isolated from *Halichondria okadai* that has potent anticancer activity, impeding mitotic division by targeting tubulin [48], is present in concentrations as low as 400 µg per kg wet weight of tissue of *Lissodendoryx* sp. [49]. Due to their low concentrations of bioactive molecules the use of these organisms poses environmental problems, because high quantities of organisms would be needed to produce enough molecules to even begin preclinical trials [50]. Yet, its structure inspired a synthetic analogue, eribulin mesylate, which is now used in breast cancer and lipocarcinoma treatment [51]. As such, sponges, molluscs, tunicates and macroalgae still remain relevant sources of new marine natural products [52].

The exploitation of other sources of bioactive marine organisms, mainly microorganisms, has also led to the discovery of new promising leads. Indeed, some of the molecules associated with macroorganisms such as sponges, may have their origin in associated microorganisms [53]. This may be the case of bryostatins, found in the marine bryozoan *Bugula neritina* [54]. The bryostatins, exemplified by bryostatin 1 (**2**) are polyketide macrolactones with neurological and anticancer properties that work by modulating the activity of the protein kinase C family and they were first isolated from this bryozoan in 1982 [55]. Yet, these macrolactones appear to be synthesised by a group of PKS genes of bacterial origin, indicating that bryostatins are produced by a bacterium [56].

Marine microorganisms, like members of Actinobacteria, Proteobacteria, Firmicutes, Cyanobacteria, fungi and dinoflagellates, have shown to be great reservoirs of bioactive molecules [52]. Yet, even though initial predictions pointed to an immediate increase in the number of natural products discovered with the shift in focus from macroorganisms to microorganisms, this did not occur [57]. However, advances in the isolation of novel taxa, provided a boost in the discovery of novel bioactive molecules. Analysis of the literature reveals an increase in the number of new molecules discovered in all microorganisms from 2014 to 2018 [52], with a special emphasis on fungi and bacteria. Members of the genus *Salinospora* (Actinobacteria) are examples of bacteria that lead to the discovery of salinosporamide A (**3**) [58]. This molecule has a potent cytotoxic activity due to a unique functionalisation of the core-fused γ-lactam-β-lactone bicyclic ring, which contributes significantly to its activity [59]. Salinosporamide A is now in phase III clinical trials for the treatment of multiple myeloma under the brand name Marizomib [60].

### 2.2. Marine Bacterial Natural Products

Although salinosporamide A is the best example of the potential present in bacteria, many novel molecular structures are discovered each year. In fact, in 2016, 179 new natural products of marine bacterial origin were discovered [61], in 2017, the number rose to 242 [62] and in 2018 a total of 240 new molecules were reported [52]. The upwards trend seen in the number of discovered molecules and the remarkable chemical diversity displayed, which ranges from peptides, siderophores and polyketides to esters, macrolactones, quinones and terpenes, shows the bacterial potential for the discovery of novel active principles.

#### 2.2.1. Antimicrobial Marine Bacterial Natural Products

Bacteria are promising sources for novel antimicrobial natural product discovery. This is primarily due to two factors. One is their variable and malleable metabolism [63] and the other is their competitive pressure for resources against other microbes. Several recent examples of natural products from marine bacteria are provided below. The novel molecule bacicyclin (**4**) (Figure 2), which is a cyclic peptide, was isolated from a *Bacillus* sp. strain BC028 isolated from the common mussel (*Mytilus edulis*) [64]. It displays antibacterial activity against *E. faecalis* and *S. aureus* with minimal inhibitory concentration (MIC) values of 8 and 12 µM, respectively, and can help in the design of analogues with increased antibiotic efficacy [64]. Anthracimycin B (**5**), a polyketide with powerful anti-Gram-positive bacteria activity that was obtained from a marine-derived *Streptomyces cyaneofuscatus* M-169, has expanded the knowledge of how the methyl group at C-2 of anthracimycins plays a role in its antibacterial effect [65]. Taromycin B (**6**), a lipodepsipeptide with potent activity against methicillin-resistant *S. aureus* and vancomycin-resistant *E. faecium* which was isolated from the marine actinomycete *Saccharomonospora* sp. CNQ-490, provides a promising start for the development of novel antibacterial scaffoldings [66]. Janthinopolyenemycin A (**7**) and B (**8**) are also polyketides and the first examples of molecules of their structural type. Janthinopolyenemycins were isolated from the proteobacterium *Janthinobacterium* spp., strains ZZ145 and ZZ148, and have activity against *Candida albicans* [67]. Streptoseomycin (**9**), a macrolactone isolated from the actinobacterium *Streptomyces seoulensis* A01, has specific activity against microaerophilic bacteria, specially the pathogen *H. pylori* [68]. This restricted activity makes streptoseomycin a good starting point for the discovery of antibiotics for the treatment of *H. pylori* infections. The polyketides ansalactams B (**10**), C (**11**) and D (**12**) are highly modified ansamycins that show weak and mild anti-methicillin-resistant *S. aureus* and were identified in cultures of *Streptomyces* sp. CNH189, isolated from marine sediments. Ansalactams B and D are cyclic polyketides with similarities to ansalactam A. However, ansalactam D shows evidence of an uncommon oxetane ring. Ansalactam C is an open polyketide chain resulting from a Baeyer–Villiger-type oxidation [69]. Micromonohalimanes A (**13**) and B (**14**) are rare halimane-type diterpenoids isolated from the actinobacterium *Micromonospora* sp. WMMC-218. Micromonohalimane A displays a very weak inhibitory effect on methicillin-resistant *S. aureus* while micromonohalimane B displays moderate bacteriostatic activity against it. Xestostreptin (**15**) is a modified diketopiperazine isolated from *Streptomyces* sp. S.4, resulting from the condensation of the aminoacids threonine and alanine [70]. Xestostreptin shows weak activity against the malarial agent *P. falciparum*. Two macrolides, branimycins B (**16**) and C (**17**) were identified from a fermentation of the actinobacterium *Pseudonocardia carboxydivorans* M-227, isolated from deep sea water [71]. Branimycin B shows moderate antibacterial activity against Gram-positive bacteria, while branimycin C displayed moderate antibacterial activity against Gram-negative bacteria. 

Utilizing a genome-assisted discovery strategy, three macrolactams, lobosamides A (**18**), B (**19**) and C (**20**), were isolated from *Micromonospora* sp. RL09-050-HVF-A [72]. Lobosamides A and B showed bioactivity against the microbial agent of African trypanosomiasis, *Trypanosoma brucei* in low concentrations. However, lobosamide C was not bioactive. Schulze and colleagues [73] also identified salinipostins A-K bicyclic phosphotriesters isolated from the actinobacterium *Salinispora* sp. RLUS08-036-SPS-B with potent and selective activity against *P. falciparum*. The salinipostin scaffold considerably differs from any of the known antimalarial compounds, representing a novel lead structure in the development of therapeutics for malaria. Experiments with salinipostin A (**21**), the most bioactive of the 11 salinipostins, indicate that it exhibits growth stage-specific effects and no apparent resistance could be identified in parasite populations. A hybrid peptide-polyketide, mollemycin A (**22**) (Figure 3) was isolated from the marine bacterium *Streptomyces* sp. CMB-M0244 and shows potent antimalarial and broad antibacterial activities [74]. Actinosporin A (**23**) is a glycosilated polyketide which shows antiparasitic activity against *T. brucei* and was isolated from a marine sponge associated *Actinokineospora* sp. EG49 [75]. Likewise, actinosporin B, was also isolated but showed no bioactivity, suggesting that actinosporin A is acting selectively against the parasite. The linear lipopetides, gageopeptides A-D (**24**–**27**), gageotetrins A–C (**28**–**30**) and gageostatins A–C (**31**–**33**) were isolated from the marine *Bacillus subtilis* 109GGC020. These lipopetides showed a range of different antimicrobial bioactivities, with gageostatins A, B and C all showing good antimicrobial activity and moderate cytotoxic activity to lung cancer cell line NCI-H23 [76]. Gageotetrins A, B and C showed potent antimicrobial bioactivities but not cytotoxic effect on human myeloid leukaemia K-562 [77]. Furthermore, gageopeptides A, B, C and D all showed good antifungal and moderate broad antibacterial activity, while not showing cytotoxicity to human myeloid leukaemia K-562 and mouse leukemic macrophage RAW 264.7 cell lines [78].

#### 2.2.2. Antiviral Marine Bacterial Natural Products

It is estimated that as many as 10^31^ viruses inhabit the oceans [79], with concentrations ranging from 3 × 10^6^ viruses mL^−1^ in deep sea waters to 10^8^ viruses mL^−1^ in coastal waters [80], many of which are bacteriophages. Consequently, marine bacteria are subjected to evolutionary pressure to develop defences against viral attacks. As such, marine bacteria may be great reservoirs of antiviral leads.

A number of examples of marine bacterial natural products with antiviral bioactivities have been recently reported. Three novel abyssomicin monomers, neoabyssomicins D (**34**) (Figure 4), E and A2 and two dimers—neoabyssomicins F (**35**) and G (**36**)—were isolated from the marine *Streptomyces koyangensis* SCSIO 5802, with neoabyssomicin D showing moderate anti-herpes simplex virus activity, and neoabyssomicins F and G showing low activity against vesicular stomatitis virus [81]. *Streptomyces* sp. OUCMDZ-3434, isolated from the marine alga *Enteromorpha prolifera*, was shown to produce five new phenolic polyketides [82]. Of these molecules, wailupemycin J (**37**) and (*R*)-wailupemycin K (**38**) proved to be bioactive against the influenza A virus (H1N1). The indolosesquiterpenoids xiamycins C (**39**), D (**40**) and E (**41**) were isolated from the marine-derived *Streptomyces* sp. #HK18, and showed strong inhibitory effect against the coronavirus porcine epidemic diarrhoea virus [83]. As such, xiamycins may provide useful leads in the development of antivirals with broader spectrum activity against other coronaviruses.

#### 2.2.3. Anticancer Marine Bacterial Natural Products

As with antimicrobials, it is ascertained that marine bacteria are great reservoirs for cytotoxic natural products. While salinosporamide A, already mentioned above, is a great example of marine cytotoxic drug discovery [58], every year, novel anticancer bioactive structures are discovered. Actinobacteria, especially those of the genera *Streptomyces* and *Micromonospora*, have been a very prolific source of cytotoxic compounds. There are several recent examples of structures isolated from these bacteria. Dentigerumycin E (**42**) (Figure 5), is a cyclic hexapeptide bearing three piperazic acids and a pyran-bearing polyketide acyl chain, isolated from the marine actinoabcterium *Streptomyces albogriseolus* JB5 [84]. It showed moderate cytotoxicity against lung carcinoma A549, colorectal cancer HCT116, breast cancer MDA-MB-231, liver cancer SK-HEP-1 and stomach cancer SNU638 cell lines, while not being cytotoxic to the normal human breast epithelial cell line MCF-10A [84]. Likewise, neothioviridamide (**43**) is a polythioamide cyclic peptide with strong cytotoxicity against human ovarian adenocarcinoma (SKOV-3), malignant pleural mesothelioma (Meso-1) and immortalized human T lymphocyte (Jurkat) cell lines [85]. It was isolated after discovery of a novel biosynthetic cluster (thioviridamide-like biosynthetic gene) in *Streptomyces* sp. MSB090213SC12 by genome mining and heterologous expression of a bacterial artificial chromosome in *Streptomyces avermitilis* SUKA. Three cyclic depsipeptides, rakicidins G-I (**44**–**46**), isolated from the marine actinobacterium *Micromonospora chalcea* FIM 02–523, have potent cytotoxic activity against the human pancreatic cancer cell line PANC-1 and human colon carcinoma cell line HCT-8 [86]. Rakicidins G-I differ in the length of their β-hydroxy fatty acid moiety. The 26-membered polyene macrolactam, FW05328-1 (**47**), isolated from *Micromonospora* sp. FIM05328, has potent bioactivity against three cells lines of human oesophageal squamous cell carcinoma (KYSE30, KYSE180 and EC109) [86]. The integration of genomic data in association with nuclear magnetic resonance (NMR) analysis allowed the determination of the stereostructure of neaumycin B, a cytovaricin-ossamycin-oligomycin macrolide, that was isolated from *Micromonospora* sp. CNY-010 [87]. Neaumycin B (**48**) has shown potent anti-human glioblastoma cell line U87 activity but it is unstable. Through genetic manipulation of promoters, six new polyketides, pactamides A–F, were isolated from *Streptomyces pactum* SCSIO 02999. Pactamides B-F showed low to moderate cytotoxic activity against human glioblastoma cell line (SF-268), human breast cancer cell (MCF-7), human large-cell lung carcinoma (NCI-H460) and human liver cancer cell (HepG2) while pactamide A (**49**) showed potent activity [88]. Two cyclodepsipeptides, streptodepsipeptides P11A (**50**) and P11B (**51**), were isolated from *Streptomyces* sp. P11-23B and displayed potent anti-proliferative bioactivity in four cell lines of human glioblastoma (U251, U87-MG, SHG-44 and C6) [89]. Research with *Streptomyces* sp. strain THS-55 yielded four new antimycin alkaloids, antimycins E-H (**52**–**55**), which showed potent cytotoxic effect on HPV-transformed human cervix adenocarcinoma (HeLa) cells and moderate anti-proliferative activity in human cervical cancer cell SiHa, human myelogenous leukaemia cell line K562, and human leukaemia cell line HL-60 [90]. However, cytotoxic effects were shown in healthy human embryonic kidney cells 293T. In a knockout mutant of *Streptomyces* sp. CHQ-64, two new alkaloids, geranylpyrrol A and piericidin F (**56**), were discovered [91]. Of these, piericidin F showed potent anti-proliferative activity against several cancer cell lines, including HeLa, human acute promyelocytic leukaemia (NB4) and human lung carcinoma (A549 and H1975).

Two new chromodepsipeptides, neo-actinomycin A (**57**) (Figure 6) and neo-actinomycin B (**58**) were isolated from a marine-derived *Streptomyces* sp. IMB094 [92]. They showed strong cytotoxic effect on human colorectal carcinoma cell line HCT116 and human lung carcinoma cell line A549.

Two new macrolides, PM100117 (**59**) and PM100118 (**60**) were isolated from a marine *Streptomyces caniferus* GUA-06-05-006A [93]. Both PM100117 and PM100118 show potent cytotoxic effect on human breast adenocarcinoma (MDA-MB-231), human lung carcinoma (A549) and human colorectal carcinoma (HT-29) cell lines. The study of a symbiotic *Streptomyces* sp. (strain 1053U.I.1a.3b) of cone snails lead to the isolation of two lobophorins, H and I [94]. Lobophorins are a large family of spirotetronates with antimicrobial and cytotoxic bioactivities. Of these, lobophorin I (**61**) showed potent cytotoxic activity against human T-cell leukaemia cell line CEM-TART. 

Besides Actinobacteria, other marine bacterial phyla are also proving to be relevant for the isolation of novel bioactive molecules. As a result of a hybrid polyketide synthase (PKS) and non-ribosomal peptide synthetase (NRPS) biosynthesis, haliamide (**62**) was isolated from a marine myxobacterium, *Haliangium ochraceum* SMP-2, and shows moderate cytotoxicity towards the HeLa cell line [95]. A novel cytotoxic indole, tetra(indol-3-yl)ethenone (**63**) was isolated from the marine proteobacterium *Pseudovibrio denitrificans* BBCC725 [96]. Tetra(indol-3-yl)ethenone has moderate cytotoxicity to human lung carcinoma cell line A549 and the mouse fibroblasts cell line L929. Another marine proteobacterium, *Labrenzia* sp. PHM005 produced a new tetrahydropyran-core polyketide and analogue of pederin [97]. This novel pederin, 18-O-demethylpederin (**64**) shows potent anti-proliferative activity against four cell lines: human lung carcinoma cell line A549, human colon adenocarcinoma cell line HT-29, human breast adenocarcinoma cell line MDA-MB-231 and human pancreas adenocarcinoma cell line PSN-1.

All these examples show the great potential displayed by marine bacteria which reveals the extraordinary chemical diversity of their metabolism. When analysing the bioactive bacteria and their taxonomic groups, Actinobacteria proved to be the most prolific and diverse producers (Table 1, Table 2 and Table 3). However, ignoring other less studied phyla denies access to valuable chemical diversity, which is essential in the drug discovery process. Moreover, marine bacterial metabolites have shown to have potential as treatment in both human and animal pathologies. Thus, marine bacterial natural products have great significance under the “One Health” framework.

## 3. Screening Methodologies for the Detection of Bioactive Marine Bacteria

### 3.1. Conventional Screening for Antimicrobials

Traditional screening methodologies for antimicrobials are based on agar diffusion assays, with the Kirby-Bauer diffusion assay being the most commonly used. This assay was standardized in the 1960s by Kirby and colleagues [98] and this and other variations of agar diffusion assays are, in fact, the models currently used by the Clinical Laboratory Standards Institute (CLSI) for antimicrobial susceptibility. The diffusion assays consist of the inoculation of specific agar media, usually Mueller-Hinton or Mueller-Hinton II agar, with the pathogenic strains under test. Paper filters impregnated with the substance to be tested are then placed on top of the medium surface and the culture is incubated usually at 37 °C, for 4 to 10 h. After this period, the medium is inspected for the formation of circular spots without bacterial growth, designated inhibition halos, and their radiuses are measured and compared to reference values of other antimicrobials, as instructed by the CLSI [99]. The Kirby-Bauer diffusion assay is very simple and relatively low-cost assay, and it allows for a rapid testing of a considerable number of pathogens and bioactive antimicrobial principles. However, it offers no information on the potential bactericidal or bacteriostatic effect produced by the molecule being tested. Moreover, diffusion in the agar medium is size-dependent, meaning that for large molecules [100], antimicrobial susceptibility may be underestimated or that false negatives can be obtained. Likewise, although there are reports of the successful use of these assays in fungi [101], not all microorganisms can be tested, as it is the case for fastidious, slow-growing and microaerophiles bacteria. Furthermore, the impossibility of its automation also hampers use of this assay in large-scale drug discovery [102].

Regarding antiparasitics, as parasites are usually obligate intracellular, they cannot be tested in agar diffusion assays. Therefore, the discovery of these drugs relies on the use of either cultures of parasites or in vivo treatments. Direct treatment of sick patients can be a useful tool, as it allows several different interactions to occur, like host related immunity or metabolism or the possibility of re-infection. However, direct treatment come with critical disadvantages, ranging from limited testing capacity to possible dangerous side effects for the patients. However, most antiparasitic therapeutics come from the veterinary world, as this saves human patients from contact with potentially toxic substances, while maintaining all the advantages of the direct treatment. Nevertheless, it can still be a costly endeavour and may yield results which do not prove to be suitable for human treatment [103]. In vitro culturing of parasites can be done with free living or infectious parasites. Free living parasites can be cultivated in axenic cultures [104] while infectious parasites come from intracellular infection of a host cell and which are more physiologically relevant [105]. By isolating the parasites from the host, the interferences caused in in vivo cultures are excluded. Yet, inactive drug precursors that must be activated by the host or drugs which activity require the synergistic effect of the host immunity cannot be tested. Moreover, in vitro testing is more resource intensive, making it hard to apply in the regions that are most affected by these diseases [103].

### 3.2. Conventional Screening for Antiviral Molecules

Vaccines, as a prophylactic approach, are the primary strategy for controlling viral outbreaks and even successfully contributing to total eradication of the agent responsible [106]. However, post-infection treatments are also needed, mostly due to the high pathogenicity that current pandemic-responsible virus displays, as stated previously [20,21,22,23]. The urgency for new and innovative antiviral therapeutics is also reinforced by the quick mutation rate of viral strains, which often leads to the creation of unique and drug-resistant viruses [107].

Successful infection by a virus is dependent on various key steps, such as the kind of entry in the host cell and the genome replication [108]. Traditional screenings for evaluating promising antiviral compounds were firstly based on non-specific approaches like the plaque inhibition assay [109], which still plays an important role in the determination of viral inhibitory concentrations. The general premise is simple, as it is based on the incubation of viral plaque-forming units in plates with overlaid layers containing media, the host cells, and the drug to be tested. The formed plaques are then stained, normally with neutral red, and counted [110]. Several modified versions of the assay have already been developed [111,112,113]. Nevertheless, antiviral screenings have progressed to more specific approaches, by focusing on the obtainment of bioactive molecules with high specificity to affect particular steps in the virus cell cycle [24]. This approach, which focuses on the obtainment of highly specific molecules, is seen in a variety of in vitro cell based and biochemical assays [108]. Reporter proteins such as luciferase or β-galactosidase can be incorporated in the assay [114,115,116].However, to do so, a virus containing a copy of a related luciferase or a-galactosidase reporter gene in its genome is needed.

As viruses will always need to be cultivated in host cells, complementary cytotoxic studies of the potential antiviral drug (detailed in the next section) are also necessary to guarantee its non-cytotoxic effect against the cell utilized in the screening.

### 3.3. Conventional Screening for Cytotoxic Molecules

The first anti-cancer therapeutics came about from in vivo experimentations, as was the case of, for example, methotrexate [117]. As already stated, these assays have disadvantages, such as toxic side effects on patients or the low number of individuals and molecules tested. The introduction of the first cultivable immortalized cancer cell lines [118] lead to a great increase in the number of discovered therapeutics, by allowing more bioactive molecules to be safely tested. In vitro cytotoxicity testing is usually performed utilizing six techniques for the evaluation of molecule bioactivity. One is staining for viable cells, as in the sulphorhodamine assay [119] and another is dye exclusion, where only dead or damaged cells are stained, as is the case of Trypan Blue [120]. Cytotoxic activity can also be assessed through metabolic activity, or a lack thereof, viability of protease activity, clonogenic capability and DNA synthesis. Methods based on metabolic activity usually rely on reduction reactions, namely the reduction of tetrazolium salts [121] and the resazurin reduction assay [122], or on the measurement of adenosine triphosphate levels, that drop as cells die [123]. In protease viability assays, cell viability is assessed by measuring the intracellular or extracellular protease activity. In intracellular protease assays, viability is positively correlated with protease activity and are performed with fluorogenic protease substrates that can penetrate cell membranes. However, in extracellular protease assays, the viability is negatively correlated with protease activity and are performed with fluorogenic protease substrates that cannot penetrate cell membranes [124]. The clonogenic capability of a cell is its ability to proliferate indefinitely, and the loss of this capacity can be related to the antitumor capacity of compounds [125]. When cells proliferate, DNA must be synthesised. In this case, quantifying the *de novo* synthesis of DNA is one of the most reliable and accurate proliferation assays. This can be done by incubation of cells with radioactive labels like ^3^H or specific thymidine analogues that can be detected though immunoblotting [126].

### 3.4. Genetic Analysis of Bioactive Potential

As mentioned earlier, the described bacterial bioactive secondary metabolites demonstrate a high degree of chemical and structural diversity and complexity. As these molecules do not appear to participate directly in the growth and development of the organism but are often important for other actions like stress response, competition or defence, they are considered secondary metabolites. Hence their potential for having biotechnologically interesting features [127]. Polyketide synthases (PKS) and non-ribosomal peptide synthetases (NRPS) are examples of families of complex enzymatic machineries that have been proven to be responsible for the production of most bacterial bioactive molecules [128]. The search for genes related to these complexes can be made using simple and low-cost molecular techniques such as DNA extraction and amplification using specific primers [129,130]. Early-on, genetic screenings can thus complement preliminary bioactivity screenings and help identify the most promising strains for use in further molecule isolation studies. In fact, this is important not only to help reduce the number of strains to work with, but also to discriminate promising strains belonging to the same species, as even phylogenetically close strains can have different genomic content regarding the presence of biosynthetic gene clusters [131]. Moreover, this approach has proven to be successful in helping select good candidates which ultimately showed interesting bioactive profiles, including the potential production of novel molecules [131,132,133]. Furthermore, with the increase in available data on sequenced bacterial genomes, genome mining techniques have rapidly evolved for detecting secondary metabolite gene clusters [134]. Several online platforms dedicated to the search for genes encoding core-biosynthetic enzymes are currently available, such as BAGEL [135], NaPDos and antiSMASH [136], amongst others. BAGEL can detect putative bacteriocins and ribosomally synthesized and post-translationally modified peptides (RiPPs) of gene clusters in bacterial genomes whereas NaPDos extracts C- and KS- domains to identify secondary metabolite genes in general. AntiSMASH is the most complete platform, as it uses a library of enzymes/protein domains that are normally seen in secondary metabolite biosynthetic pathways to identify possible hits with 44 different types of gene clusters [136]. These include, not only various types of PKS, NRPS and PKS/NRPS hybrids, but also other important secondary metabolite gene clusters for biosynthesis of compounds like RiPPs, terpenoids/isoprenoids, and ectoines, among others. This approach is already successfully assisting in discovering and characterize new natural products, as is the case for the macrolactams lobosamide A, B and C [72], which were characterized with the help of antiSMASH and the polythioamide neothioviridamide [85], whose biosynthetic cluster belongs to a cryptic RiPP and was heterogeneously expressed in *S. avermitilis* (discussed in Section 2.2.1 and Section 2.2.3). Genome mining for secondary metabolite genes can also be applied in metagenomic data of still uncultured organisms, as well as to help unveil the biotechnological potential of unique and still under-explored bacterial groups, such as, for example, the planctomycetes [137,138,139].

### 3.5. High-Throughput Screening Methodologies

Because of the drawbacks imposed by traditional screening methodologies, biotechnological pharmaceutical companies have largely abandoned these assay models in favour of high-throughput screening (HTS) methodologies. HTS methodologies are characterized by rapid, high-volume testing of compounds, in the range of 10^5^ to 10^6^ molecules per day, and are inserted in a whole drug discovery strategy directed towards novel therapeutic leads [140]. HTS are easily miniaturized in 96-, 384- or 1,536-microwell plate formats and are generally based on cell or biochemical targets in which the response is measured as either an increase or a decrease of a signal. The signal measured is usually based on absorbance methods, e.g., fluorescence [141]. Non-conventional methodologies, genetic tools and HTS methodologies, are less labour demanding, improve the speed and capacity of testing, and discovering of novel bioactive molecules. They potentially allow the decrease of the needed time for obtainment of new pharmaceuticals, which is a fundamental aspect in the lengthy process of drug discovery.

## 4. Strategies for Avoiding Reisolation and Recharacterization of Known Bioactive Compounds

The great chemical diversity present in natural molecules has been the impetus behind the discovery of novel therapeutics [142]. Yet, the isolation and characterization of novel natural molecules is a laborious, time consuming endeavour that requires particular expertise and competences [143]. This is mainly due to the presence of known and redundant compounds in crude extracts [144]. Thus, a strategy for recognizing and eliminating already known bioactive natural molecules in the early stage of the screening process is crucial. This process is called dereplication [145].

The dereplication process analyses previously identified active samples in preliminary biological screenings through a combination of analytical and spectroscopic methods to putatively identify any possible known bioactive molecules present. In this section, current strategies for the dereplication of bioactive extracts will be explained.

### 4.1. Analytical Separation Techniques

Crude bacterial extracts are complex mixtures of organic molecules [146], usually containing nutrients, primary and secondary metabolites [147]. Due to this complexity, in the dereplication process, crude extracts must first be fractionated prior to any spectroscopic analysis [145]. While many developed separation techniques exist, like thin-layer chromatography, gas chromatography and capillary electrophoresis to name but a few, the most commonly used techniques in the dereplication process pivot between high-performance liquid chromatography (HPLC) and ultra-high-performance liquid chromatography (UHPLC) [148]. The extensive use of HPLC systems in the isolation of natural products has allowed it to become a powerful and versatile tool, as HPLC systems can be easily hyphenated with detection techniques (like ultraviolet/visible (UV/Vis) detectors, mass (MS) and NMR spectrometers, see below) and copious amounts of data available, makes HPLC the *de facto* standard chromatographic technique for dereplication [142]. Still, while UHPLC systems are less used, they can operate at very high pressures with very small particle size columns, which grants them great sensitivity while diminishing sample and eluent volumes used and the analysis time [149]. However, both techniques allow for the use of compatible column types and chemical phases. Because of the chemical nature of most natural products, small molecules with molecular weights bellow 1000 Da and low or no polarity [147], the best method for their separation is revered-phase chromatography which allows a longer retention time in the column during the polar and elution in the non-polar phase [150,151]. The time that the metabolite is retained in the column, also called retention time, gives an important chromatographic property intrinsic to each molecule that can be used in its identification [145,148,152].

### 4.2. Detection Methods

In the dereplication process, molecules are identified due to intrinsic properties, like absorption peaks or molecular mass, which can be measured by a number of detection techniques. The most common techniques include, UV/Vis diode array detectors (DAD), mass spectrometers and NMR spectrometers [142].

In UV/Vis spectroscopy, UV/vis spectra can be immediately acquired as part of the chromatographic separation of the extracts [147]. UV/vis spectra are normally obtained through the use of DAD, which can scan a range of wavelengths and produce the absorption spectrum. UV/Vis spectroscopy gives useful information on the molecules, such as chromophores present and the max absorption wavelength (λ_max_), giving important structural information for the identification of yet unknown molecules [153]. Yet, but for the dereplication process, the full spectrum is a superior and more definitive approach, as it contains more information than just the λ_max_ can give [148]. However, in the dereplication process, UV/vis spectra are normally used only as a secondary criterion. Currently, the principal criterion for identification is based on high resolution-MS (HRMS) [142]. MS is a destructive technique that provides the mass-to-charge ratio (*m/z*) of the ionized molecule or ionized fragments of the molecule. MS data is presented as spectrum of *m/z* and relative intensity. The *m/z* can be used to estimate the molecular mass of the molecule and thus, its molecular formula [154]. Additionally, as the ionization of the molecule may fragment it in smaller pieces, this fragmentation pattern can also be used in the identification of the molecule. Moreover, if it is a novel molecule, the fragmentation pattern can give useful information for structural elucidation of the molecule [155]. Initial dereplication efforts were based around low resolution-MS (LRMS) [156]. However, LRMS is ineffective in resolving between two very closely weight molecules, as LRMS can only confidently detect up to the integer of the mass. Thus, HRMS, which can give extremely accurate molecular masses [154], was adopted. Nevertheless, an LC-DAD-MS dereplication only putatively identifies the compounds present in the extract, because of the possible existence of regioisomers or stereoisomers [148]. Therefore, full confirmation of the identity is only possible through NMR spectroscopy. NMR spectroscopy of natural products is normally performed on protons (^1^H-NMR) and carbons (^13^C-NMR) where the chemical shifts measured correlate with the position and type of each atom in the molecule. NMR has various advantages compared to MS, as it can analyse all kinds of small metabolites and samples can be recovered for further analyses. It is considered the universal detector for structure elucidation, has high analytical reproducibility and simplicity of sample preparation [142] and can be used for checking the purity of the sample, a problem that can arise from the common purification steps undertaken in most natural product isolation [145]. An example of this are lipids, problematic impurities that are undetectable in LC-DAD-MS analysis because they have low UV absorption, have difficulty of ionizing and high lipophilicity, which, ^1^H-NMR can easily recognize by the presence of the large peak of the lipidic methylene chain [157]. Yet, for NMR spectroscopy, sample size is a limiting factor, as it may require up to milligrams of the sample [142]. In contrast, MS can work with attomolar concentrations of sample [158]. This limitation in NMR spectroscopy is especially important in the dereplication process, as crude extracts may only contain minute quantities of molecules.

### 4.3. Natural Product Databases 

Databases of natural products are critical to successfully complete the dereplication process [142]. Databases contain crucial information on natural products, including molecular weight, exact mass and fragmentation pattern, UV/vis λ_max_ absorption or full spectra and NMR spectra or chemical shifts [148]. Databases of natural products should also contain relevant taxonomic information of the producer organism and known biological activity. For the purpose of identification of natural products, public domain and commercial databases are available. The most important commercial databases include Chemical Abstracts Service’s Registry File (CAS) [159], NaprAlert [160], the Chapman & Hall Dictionary of Natural Products (DNP), AntiBase [161], MarinLit [162], the open-access databases Global Natural Products Social Networking (GNPS) [163,164], The Natural Products Atlas (NPAtlas) [165,166] and Small Molecule Accurate Recognition Technology (SMART) [167,168]. The CAS database comprises the largest online repository of chemical structures, including natural products. NaprAlert has significant resources for dereplicating the terrestrial sources. MarinLit is a database dedicated to marine natural products established by Professors John Blunt and Murray Munro. Records contain bibliographic information and the database can be searched by querying substructure, ^13^C- and ^1^H-NMR shift data, exact mass, chemical formula, or UV λ_max_ absorption. AntiBase focuses on molecules from microorganisms and higher fungi. AntiBase incorporates molecular formula and mass, melting point, optical rotation, UV λ_max_ absorption, ^13^C and ^1^H-NMR shift data and mass spectra. AntiBase also contains bioactivity data and information on origin, isolation and literature sources.

The DNP is considered the most complete natural product database, containing relevant bibliographic references [148]. The DNP contains information on exact mass and molecular formula, UV λ_max_ absorption, biological sources and activity. Of the commercial databases, for marine bacterial natural products, MarinLit, AntiBase and the DNP and its subset on marine natural products (DMNP) are considered the most relevant databases for marine bacterial natural products dereplication. GNPS is an open-access database hosted at University of California, San Diego of LC and MS/MS spectra below *m/z* 2000 of natural products with data of over 18,000 compounds. GNPS also can detect sets of spectra from related molecules, even if the spectra used is not matched to any known compounds [142]. It allows the use of experimental MS/MS spectra as input in searches, allowing the direct dereplication of molecules present in extracts or chromatographic fractions without the need for determining the molecular formulae of their components. Additionally, the algorithm behind the platform allows networks of molecules based on their cosine similarity to be established, a process that has proved to be very useful in the identification of new members of known families of natural products. Bioactivity data can also be added to the process, allowing the direct correlation between the presence of certain components in samples and their bioactivity, using the so-called “bioactive molecular networking” (BMN) approach [169]. 

GNPS molecular networking has been successfully applied to the discovery of several bioactive natural products, including the antimicrobial angucycline-derived polycyclic aromatic polyketide lugdunomycin (**65**) (Figure 7) [170], and the aromatic polyketide accramycin A (**66**) [171], both isolated from *Streptomyces* sp., as well as the antibacterial chlorinated cyclic hexapeptides noursamycins A-F (**67-72**), obtained from cultures of *Streptomyces noursei* NTR-SR4 [172]. NPAtlas is a database which includes over 25,500 compounds from microbial sources. It contains referenced data for structure, compound names, source organisms, isolation references, total syntheses, and instances of structural reassignment and is integrated with other natural product databases, like GNPS [165]. SMART is an NMR database [in particular, non-uniform sampling heteronuclear single quantum coherence (NUS HSQC)] assisted by a deep convolutional neural network (dCNN) to not only identify compounds, but for novel unidentified ones, placing them in natural product families with similar HSQC spectra, thus allowing for faster structural elucidation. As SMART is assisted by dCNN, as the size of the training set increases for a given compound class, so does its accuracy and capability of prediction of a new compound’s structure improve [167]. Additionally, with further training, SMART has the potential to enable novel experimentation at the chemistry-biology interface, like predictions of biological activity and or target. The SMART technology in combination with GNPS was successfully applied recently to the discovery of symplocolide A (**73**), a new chimeric swinholide-like macrolide with cytotoxic properties against the NCI-H460 human lung cancer cell line obtained from extracts of the filamentous marine cyanobacterium *Symploca* sp. [173].

## 5. Conclusions

It is evident that great heath challenges are still in need of novel pharmaceutical responses. Resistance of bacteria and fungi to antibiotics, of parasites like malaria to the available treatments, unavailability of medicines to treat new viruses and efficient drugs for cancer treatment, require additional research and development focus in discovering novel bioactive molecules. Oceans, home to a substantial portion of the world’s biodiversity, are still underexplored, and are an important source of drugs and drug leads. Since 2014, several novel marine bacterial natural products, obtained mainly from Actinobacteria, Firmicutes and Proteobacteria, were described and reported in this review. In supplement to conventional methods, high-throughput analysis, like genome and metagenomic analysis and HTS, and the use of new dereplication and identification tools, will foster the discovery of new leads. This review shows that marine bacteria are key to the development of new pharmaceuticals, especially if combined with a rational, high-throughput approach.

## Figures and Tables

**Figure 1 antibiotics-09-00455-f001:**
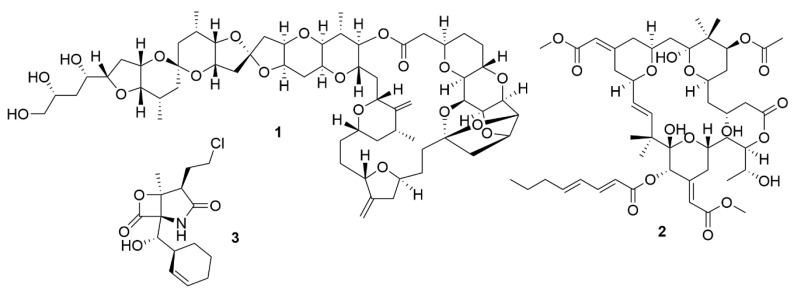
Bioactive metabolites isolated from marine organisms. Halichondrin B (**1**), a macrolide first isolated from *H. okadai* but also present in other sponges. Bryostatin 1 (**2**), which belongs to a family of polyketide macrolides first identified in the marine bryozoan *B. neritina*. Salinosporamide A (**3**), is a proteasome inhibitor isolated from bacteria from the genus *Salinospora* and is in phase III clinical trials for the treatment of multiple myeloma.

**Figure 2 antibiotics-09-00455-f002:**
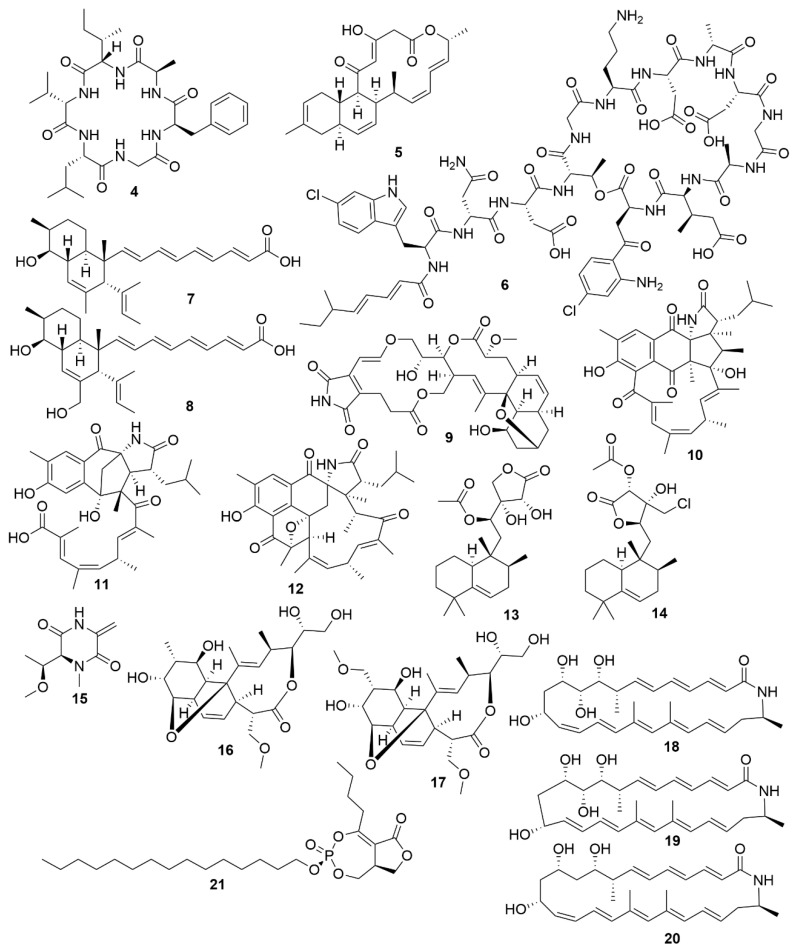
Examples of recently isolated antimicrobial natural products from marine bacteria. Bacicyclin (**4**), a cyclic peptide isolated from a *Bacillus* sp. BC028. Anthramicin B (**5**), a polyketide isolated from *S. cyaneofuscatus* M-169. Taromycin B (**6**), a lipopeptide from *Saccharomonospora* sp. CNQ-490. Janthinopolyenemycin A (**7**) and Janthinopolyenemycin B (**8**). The janthinopolyenemycins are polyketides isolated from two strains of the genus *Janthinobacterium*. Streptoseomycin (**9**), a macrolactone isolated from *S. seoulensis*. Ansalactam B (**10**), a pentacyclic polyketide. Ansalactam C (**11**), an open polyketide unlike ansalactam B. Ansalactam D (**12**), a hexacyclic polyketide. Ansalactams B, C and D were isolated from *Streptomyces* sp. CNH189. Micromonohalimane A (**13**). Micromonohalimane B (**14**). Micromonohalimanes A and B are terpenes isolated from *Micromonospora* sp. WMMC-218. Xestostreptin (**15**), a diketopiperazine isolated from *Streptomyces* sp. S.4. Branimycin B (**16**) and Branimycin C (**17**) C are macrolides isolated from the deep-sea bacterium *P. carboxydivorans* M-227. Lobosamide A (**18**), Lobosamide B (**19**) and Lobosamide C (**20**) are macrolactams isolated from *Micromonospora* sp. RL09-050-HVF-A. Salinipostin A (**21**), a bicyclic phosphotriester isolated from *Salinispora* sp. RLUS08-036-SPS-B.

**Figure 3 antibiotics-09-00455-f003:**
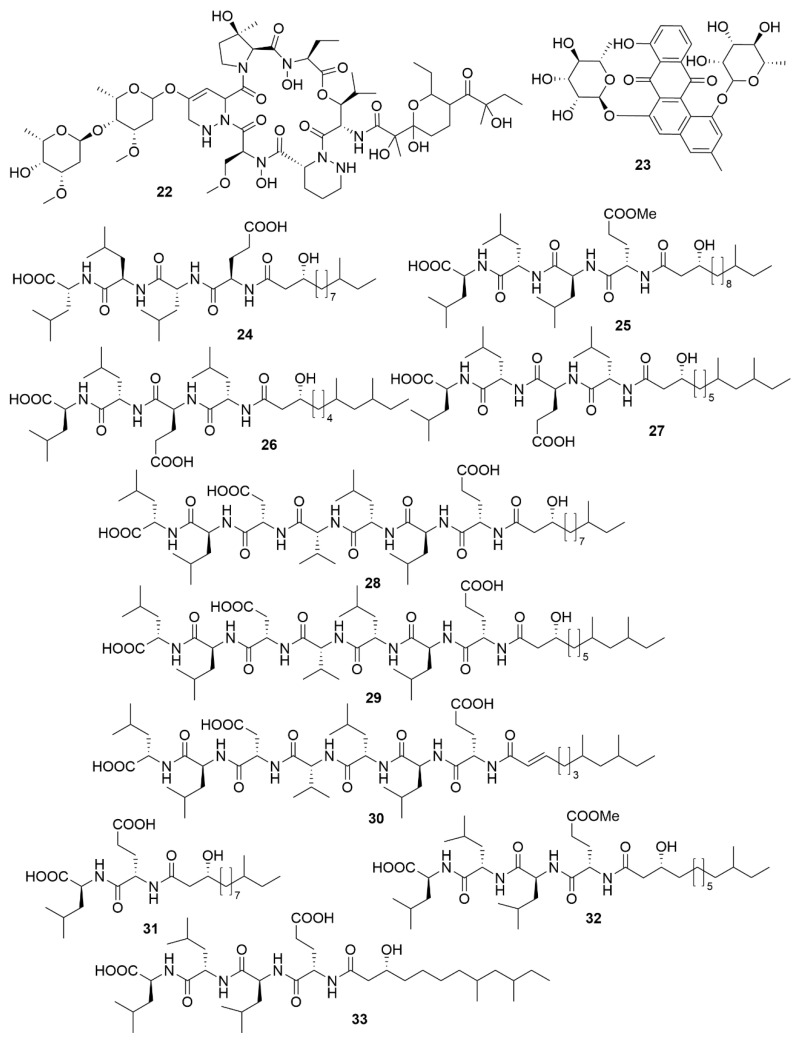
Examples of recently isolated antimicrobial natural products from marine bacteria. Mollemycin A (**22**) is a hybrid peptide-polyketide isolated from *Streptomyces* sp. CMB-M0244. Actinosporin A (**23**), a polyketide isolated from *Actinokineospora* sp. EG49. Gageopeptide A (**24**). Gageopeptide B (**25**). Gageopeptide C (**26**), Gageopeptide D (**27**), Gageotetrin A (**28**). Gageotetrin B (**29**). Gageotetrin C (**30**). Gageostatin A (**31**). Gageostatin B (**32**). Gageostatin C (**33**). The gageopeptides, gageotetrins and gageostatins are linear lipopeptides isolated from *B. subtilis* 109GGC020.

**Figure 4 antibiotics-09-00455-f004:**
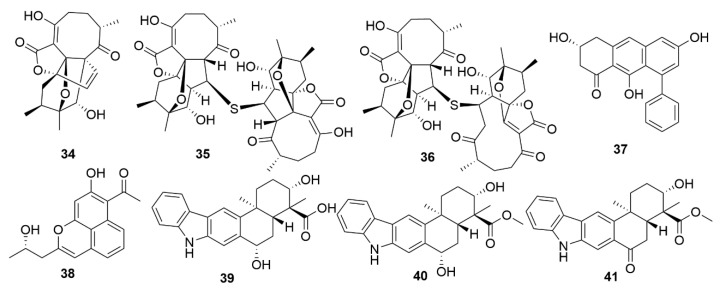
Examples of recently isolated natural products from marine bacteria with antiviral activity. Neoabyssomicin D (**34**). Neoabyssomicin F (**35**) and Neoabyssomicin E (**36**) are polycyclic polyketides isolated from *S. koyangensis* SCSIO 5802. Wailupemycin J (**37**) and (*R*)-wailupemycin K (**38**) are phenolic polyketides isolated from *Streptomyces* sp. OUCMDZ-3434. Xiamycin C (**39**), Xiamycin D (**40**) and Xiamycin E (**41**) were isolated from *Streptomyces* sp. #HK18.

**Figure 5 antibiotics-09-00455-f005:**
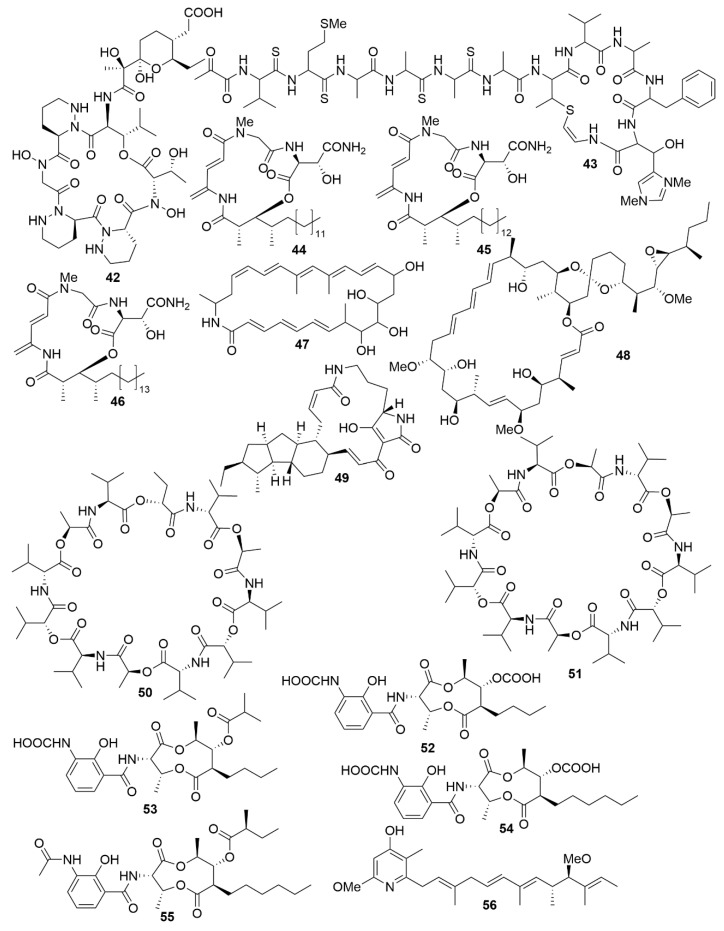
Examples of recently isolated cytotoxic natural products from marine bacteria. Dentigerumycin E (**42**), a cyclic hexapeptide isolated from *S. albogriseolus* JB5. Neothioviridamide (**43**), a cyclic peptide from *Streptomyces* sp. MSB090213SC12. Rakicidin G (**44**), Rakicidin H (**45**) and Rakicidin I (**46**) are cyclic depsipeptides isolated from *M. chalcea* FIM 02–523. FW05328-1 (**47**), a polyene macrolactam isolated from *Micromonospora* sp FIM05328. Neaumycin B (**48**), a macrolide from *Micromonospora* sp. CNY-010. Pactamide A (**49**), a polyketide isolated from *S. pactum* SCSIO 02999. Streptodepsipeptide P11A (**50**) and Streptodepsipeptide P11B (**51**) are cyclodepsipeptides from *Streptomyces* sp. P11-23B. Antimycin E (**52**), Antimycin F (**53**), Antimycin G (**54**) and Antimycin H (**55**) are alkaloids isolated from *Streptomyces* sp. THS-55. Piericidin F (**56**) is an alkaloid isolated *Streptomyces* sp. CHQ-64.

**Figure 6 antibiotics-09-00455-f006:**
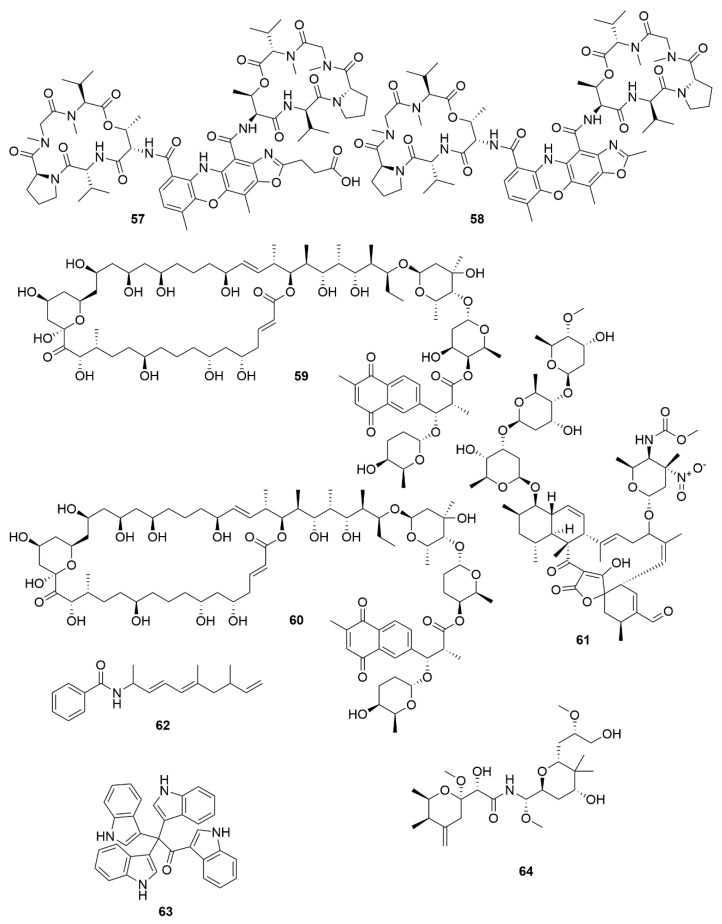
Examples of recently isolated cytotoxic natural products from marine bacteria. Neo-actinomycin A (**57**) and Neo-actinomycin B (**58**) are chromopeptides from *Streptomyces* sp. IMB094. PM100117 (**59**) and PM100118 (**60**) are macrolides isolated from *S. caniferus* GUA-06-05-006A. Lobophorin I (**61**), a spirotetronate isolated from *Streptomyces* sp. 1053U.I.1a.3b. Haliamide (**62**), a hybrid of a polyketide synthase from *H. ochraceum* SMP-2. Tetra(indol-3-yl)ethenone (**63**), an indole isolated from *P. denitrificans* BBCC725. O-Demethylpederin (**64**), a polyketide with a tetrahydropyran-core from *Labrenzia* sp. PHM005.

**Figure 7 antibiotics-09-00455-f007:**
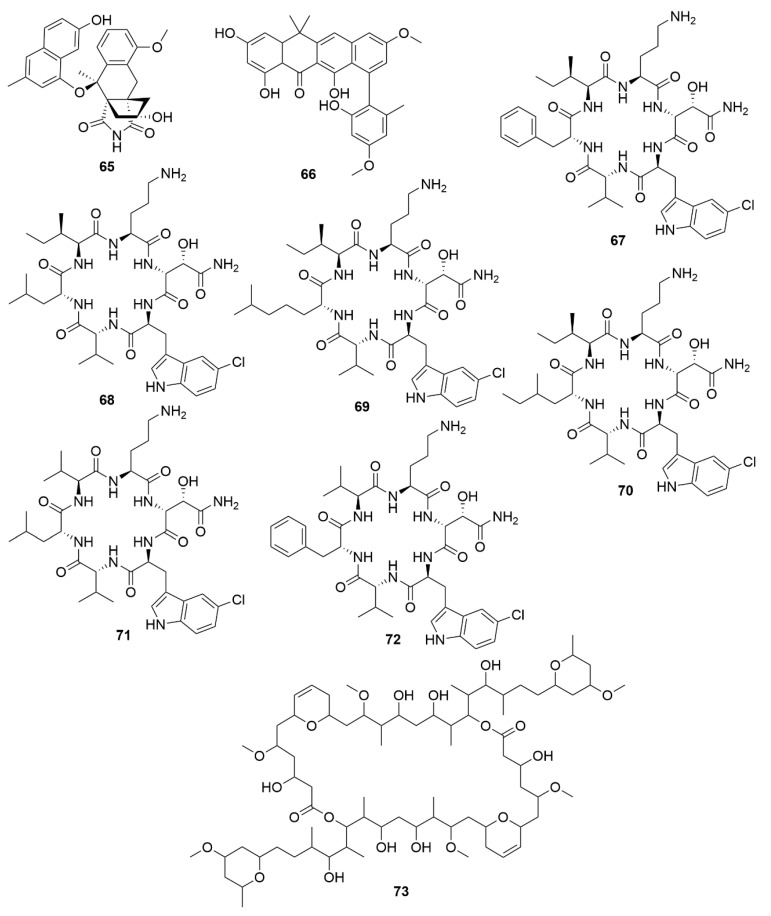
Examples of recently isolated bacterial natural products employing the GNPS and SMART technologies. Lugdunomycin (**65**), a novel angucycline-derived polycyclic aromatic polyketide with antimicrobial properties, isolated from *Streptomyces* sp. QL37 utilising a metabolomic combined with MS-based molecular networking analysis. Accramycin A (**66**), a naphthacene-type aromatic polyketide isolated from *Streptomyces* sp. MA37, which characterization was aided by MS/MS molecular networking. Noursamycin A (**67**), Noursamycin B (**68**), Noursamycin C1 (**69**), Noursamycin C2 (**70**), Noursamycin D (**71**) and Noursamycin E (**72**) are chlorinated cyclic hexapeptides isolated from *S. noursei* with the aid of MS/MS molecular networking. Symplocolide A (**73**), a swinholide-like macrolide with cytotoxic properties obtained from extracts of the filamentous marine cyanobacterium *Symploca* sp. by employing both SMART technologies and MS/MS molecular networking.

**Table 1 antibiotics-09-00455-t001:** Recently isolated bioactive molecules from marine bacteria with antimicrobial properties.

Molecule	Bacterial Origin		Chemical Structure	Bioactivity	
	Strain Identification	Phyla		Effect	Target
Actinosporin A	*Actinokineospora* sp. EG49.	Actinobacteria	Polyketide	AP	TBB
Lobosamide A	*Micromonospora* sp. RL09-050-HVF-A	Actinobacteria	Polyketide	AP	TBB
Lobosamide B	*Micromonospora* sp. RL09-050-HVF-A	Actinobacteria	Polyketide	AP	TBB
Micromonohalimane A	*Micromonospora* sp. WMMC-218	Actinobacteria	Polyketide	AB	MRSA
Micromonohalimane B	*Micromonospora* sp. WMMC-218	Actinobacteria	Polyketide	AB	MRSA
Branimycin B	*Pseudonocardia carboxydivorans* M-227	Actinobacteria	Polyketide	AB	G+
Branimycin C	*Pseudonocardia carboxydivorans* M-227	Actinobacteria	Polyketide	AB	G-
Taromycin B	*Saccharomonospora* sp. CNQ-490	Actinobacteria	Peptide	AB	MRSA; VRE
Salinipostin A	*Salinispora* sp. RLUS08-036-SPS-B	Actinobacteria	Polyketide	AP	PF
Anthracimycin B	*Streptomyces cyaneofuscatus* M-169	Actinobacteria	Polyketide	AB	G+
Streptoseomycin	*Streptomyces seoulensis* A01	Actinobacteria	Polyketide	AB	H. pylori
Mollemycin A	*Streptomyces* sp. CMB-M0244	Actinobacteria	Peptide-polyketide	AB/AP	Broad spectrum/PF
Ansalactam B	*Streptomyces* sp. CNH189	Actinobacteria	Polyketide	AB	MRSA
Ansalactam C	*Streptomyces* sp. CNH189	Actinobacteria	Polyketide	AB	MRSA
Ansalactam D	*Streptomyces* sp. CNH189	Actinobacteria	Polyketide	AB	MRSA
Xestostreptin	*Streptomyces* sp. S.4	Actinobacteria	Peptide	AP	PF
Bacicyclin	*Bacillus* sp. BC028	Firmicutes	Peptide	AB	G+
Gageopeptide A	*Bacillus subtillis* strain 109GGC020	Firmicutes	Peptide	AB/AF	Broad spectrum
Gageopeptide B	*Bacillus subtillis* strain 109GGC020	Firmicutes	Peptide	AB/AF	Broad spectrum
Gageopeptide C	*Bacillus subtillis* strain 109GGC020	Firmicutes	Peptide	AB/AF	Broad spectrum
Gageopeptide D	*Bacillus subtillis* strain 109GGC020	Firmicutes	Peptide	AB/AF	Broad spectrum
Gageotetrin A	*Bacillus subtillis* strain 109GGC020	Firmicutes	Peptide	AF	Broad spectrum
Gageotetrin B	*Bacillus subtillis* strain 109GGC020	Firmicutes	Peptide	AF	Broad spectrum
Gageotetrin C	*Bacillus subtillis* strain 109GGC020	Firmicutes	Peptide	AF	Broad spectrum
Gageostatin A	*Bacillus subtillis* strain 109GGC020	Firmicutes	Peptide	AB/AF	Broad spectrum
Gageostatin B	*Bacillus subtillis* strain 109GGC020	Firmicutes	Peptide	AB/AF	Broad spectrum
Gageostatin C	*Bacillus subtillis* strain 109GGC020	Firmicutes	Peptide	AB/AF	Broad spectrum
Janthinopolyenemycin A	*Janthinobacterium* spp. ZZ145 and ZZ148	Proteobacteria	Polyketide	AF	CA
Janthinopolyenemycin B	*Janthinobacterium spp.* ZZ145 and ZZ148	Proteobacteria	Polyketide	AF	CA

AB = Antibacterial, AF = Antifungal AP = Antiparasitic; CA = *Candida albicans*; G+ = Gram-positive bacteria; G- = Gram-negative bacteria; MRSA = Methicillin-resistant *S. aureus*; PF = *Plasmodium falciparum*; TBB = *Trypanosoma brucei;* VRE = Vancomycin-resistant *E. faecium.*

**Table 2 antibiotics-09-00455-t002:** Recently isolated bioactive molecules from marine bacteria with antiviral properties.

Molecule	Bacterial Origin		Chemical Structure	Bioactivity
	Strain Identification	Phyla		Target
Neoabyssomicin D	*S. koyangensis* SCSIO 5802	Actinobacteria	Polyketide	HSV
Neoabyssomicin F	*S. koyangensis* SCSIO 5802	Actinobacteria	Polyketide	VSV
Neoabyssomicin G	*S. koyangensis* SCSIO 5802	Actinobacteria	Polyketide	VSV
Wailupemycin J	*Streptomyces* sp. OUCMDZ-3434	Actinobacteria	Polyketide	H1N1
R-wailupemycin K	*Streptomyces* sp. OUCMDZ-3435	Actinobacteria	Polyketide	H1N1
Xiamycin C	*Streptomyces* sp. #HK18	Actinobacteria	Polyketide	PEDV
Xiamycin D	*Streptomyces* sp. #HK18	Actinobacteria	Polyketide	PEDV
Xiamycin E	*Streptomyces* sp. #HK18	Actinobacteria	Polyketide	PEDV

HSV = Herpes simplex virus; VSV = vesicular stomatitis virus; H1N1 = influenza A virus; PEDV = porcine epidemic diarrhea virus.

**Table 3 antibiotics-09-00455-t003:** Recently isolated bioactive molecules from marine bacteria with cytotoxic properties.

Molecule	Bacterial Origin		Chemical Structure	Bioactivity
	Strain Identification	Phyla		Target
Dentigerumycin E	*Streptomyces albogriseolus* JB5	Actinobacteria	Peptide	HCT116; A549; MDA-MB-231; SK-HEP-1; SNU638
Neothioviridamide	*Streptomyces* sp. MSB090213SC12	Actinobacteria	Peptide	SKOV-3; Meso-1; Jurkat
Rakicidin G	*Micromonospora chalcea* FIM 02-523	Actinobacteria	Peptide	PANC-1; HCT-8
Rakicidin H	*Micromonospora chalcea* FIM 02-523	Actinobacteria	Peptide	PANC-1; HCT-8
Rakicidin I	*Micromonospora chalcea* FIM 02-523	Actinobacteria	Peptide	PANC-1; HCT-8
FW05328-1	*Micromonospora* sp. FIM05328	Actinobacteria	Polyketide	KYSE30; KYSE180; EC109
Neaumycin B	*Micromonospora* sp. CNY-010	Actinobacteria	Polyketide	U87
Pactamide A	*Streptomyces pactum* SCSIO 02999	Actinobacteria	Polyketide	SF-268; MCF-7; NCI-H460; Hep-G2
Pactamide B	*Streptomyces pactum* SCSIO 02999	Actinobacteria	Polyketide	SF-268; MCF-7; NCI-H460; Hep-G2
Pactamide C	*Streptoyces pactum* SCSIO 02999	Actinobacteria	Polyketide	SF-268; MCF-7; NCI-H460; Hep-G2
Pactamide D	*Streptomyces pactum* SCSIO 02999	Actinobacteria	Polyketide	SF-268; MCF-7; NCI-H460; Hep-G2
Pactamide E	*Streptomyces pactum* SCSIO 02999	Actinobacteria	Polyketide	SF-268; MCF-7; NCI-H460; Hep-G2
Pactamide F	*Streptomyces pactum* SCSIO 02999	Actinobacteria	Polyketide	SF-268; MCF-7; NCI-H460; Hep-G2
Streptodepsipeptide P11A	*Streptomyces* sp. P11-23B	Actinobacteria	Peptide	U251; U87; SHG-44; C6
Streptodepsipeptide P11B	*Streptomyces* sp. P11-23B	Actinobacteria	Peptide	U251; U87; SHG-44; C6
Antimycin E	*Streptomyces* sp. THS-55	Actinobacteria	Polyketide	HeLa; SiHa; K562; HL-60; 293T
Antimycin F	*Streptomyces* sp. THS-55	Actinobacteria	Polyketide	HeLa; SiHa; K562; HL-60; 293T
Antimycin G	*Streptomyces* sp. THS-55	Actinobacteria	Polyketide	HeLa; SiHa; K562; HL-60; 293T
Antimycin H	*Streptomyces* sp. THS-55	Actinobacteria	Polyketide	HeLa; SiHa; K562; HL-60; 293T
Piericidin F	*Streptomyces* sp. CHQ-64	Actinobacteria	Polyketide	HeLa; NB4; A549; H1975
Neo-actinomycin A	*Streptomyces* sp. IMB094	Actinobacteria	Peptide	HCT116; A549
Neo-actinomycin B	*Streptomyces* sp. IMB094	Actinobacteria	Peptide	HCT116; A549
PM100117	*Streptomyces caniferus* GUA-06-05-006A	Actinobacteria	Polyketide	A549; MDA-MB-231; HT-29
PM100118	*Streptomyces caniferus* GUA-06-05-006A	Actinobacteria	Polyketide	A549; MDA-MB-231; HT-29
Lobophorin I	*Streptomyces* sp. 1053U.I.1a.3b	Actinobacteria	Polyketide	CEM-TART
Haliamide	*Haliangium ochraceum* SMP-2	Myxobacteria	Polyketide	HeLa
Tetra(indol-3-yl)ethanone	*Pseudovibrio denitrificans* BBCC725	Proteobacteria	Polyketide	L929; A549
18-O-demethylpederin	*Labrenzia* sp. PHM005	Proteobacteria	Polyketide	A549; HT-29; MDA-MB-231; PSN-1;

HCT116 = HCT-8 = HT-29 = human colorectal carcinoma; A549 = H1975 = human lung carcinoma; MDA-MB-231 = MCF-7 = human breast adenocarcinoma; SK-HEP-1 = Hep-G2 = human hepatic adenocarcinoma; SNU638 = human gastric carcinoma; SKOV-3 = human ovarian adenocarcinoma; Meso-1 = malignant pleural mesothelioma; Jurkat = immortalized human T lymphocyte; PANC-1 = PSN-1 = human pancreas adenocarcinoma; KYSE30 = KYSE180 = EC109 = human oesophageal squamous cell carcinoma; U251 = U87 = SHG-44 = C6 = SF-268 = human glioblastoma cell line; HeLa = SiHa = human cervix adenocarcinoma; K562 = HL-60 = NB4 = human leukaemia cell line; 293T = human embryonic kidney cells; L929 = mouse fibroblasts cell; CEM-TART = human T-cell leukaemia.

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
