# Peer review of "From Ocean to Medicine: Pharmaceutical Applications of Metabolites from Marine Bacteria"

_antibiotics, 2020, doi:10.3390/antibiotics9080455_

Round 1

Reviewer 1 Report

This is a well-written review paper discussing bacterial derived natural products as tools to fight them. Furthermore, novel molecules, from the identification of bioactive bacterial crude extracts to the isolation and chemical characterization of these molecules within the
 the framework of the One Health approach are discussed.

The review states global information on the subject as it explores involved organisms,economic burden, marine microbiology, and chemistry aspects.

It is an original subject. The bibliography is up to date and correctly discussed.

My suggestion is to publish the paper in its current form as it should be of high interest to readers.

Highlights

As stated this is scientifically sound review as it explores all different aspects of novel natural products issued from marine bacteria and their potential in the treatment of bacterial, viral, parasitic infections and cancer. In the first part the article states the important public health problems and cancer and their economic burden for the society. In tables which facilitate the lecture, are shown the major antimicrobial natural products discovered from marine bacteria and the Development of bioactive molecules having antiviral, antibacterial and anticancer potential. Moreover, the authors discussed on the detection methods; conventional methodologies but also high-throughput analysis, like genome analysis and HTS and make a critical review on the strategies to follow for recognizing and eliminating characterization of already discovered bioactive natural products through a combination of analytical and spectroscopic methods in the early stage of the screening process. They states also the importance of Database of natural products fully characterized for avoiding dereplication.

I found that this complete and in depth review study of the subject and it must be published.

Weakness

The authors must correct on line 48-49 in italics

Proteus sp

Also sp  after bacterial names must be in italics

Author Response

We thank the reviewer for all the corrections pointed out.

"The authors must correct on line 48-49 in italics

R: Done

Proteus sp

R: Done

Also sp  after bacterial names must be in italics

R: We are sorry, but this is not the correct code for species designation. sp is not in italics"

Reviewer 2 Report

The authors proposed a review on a theme already well explored in the scientific literature. The authors did not present a plausible justification of the need for another review article in this area. The authors should point out the novelty (i.e. the added value for readers) of this review article.

Authors have just covered antimicrobial, antiviral and anticancer compounds; hence the title should be changed accordingly.

When talking about antibiotic resistance, it would be nice for the readers to see the timeline of drug discovery and drug resistance development throughout the years.

Also, non-conventional strategies for the discovery of natural products with potential bioactivity should be discussed.

While discussing about dereplication strategy under section 4, more compounds should be reviewed in terms of application of these strategies. Similarly, compounds isolated using genomic and metabolomic mining strategy should be reviewed in more details rather than just explaining the databases.

Kindly reorganize the structures in the continuous number format rather than alphabets separate for each figure.

The introduction paragraph is very long and need to be reorganized. It lacks the logical flow.

Timeliness is an essential factor for review articles. I would like to see the timeframe this review article covers.

Author Response

We thank the reviewer for pointing out some of the weaknesses in the manuscript. Bellow are our answers to points raised by the reviewer (in bold):
The authors proposed a review on a theme already well explored in the scientific literature. The authors did not present a plausible justification of the need for another review article in this area. The authors should point out the novelty (i.e. the added value for readers) of this review article.

Authors have just covered antimicrobial, antiviral and anticancer compounds; hence the title should be changed accordingly.

R: We acknowledge the reviewer suggestion and inserted the word “some” in the title:

“From ocean to medicine: some pharmaceutical applications of metabolites from marine bacteria”

When talking about antibiotic resistance, it would be nice for the readers to see the timeline of drug discovery and drug resistance development throughout the years.

R: The aim of this review is focused on marine bacterial natural products, evidencing the most updated achievements. Thus, we consider not relevant the inclusion of a timeline of drug discovery and drug resistance development

Also, non-conventional strategies for the discovery of natural products with potential bioactivity should be discussed.

R: As suggested, we improved the discussion of non-conventional strategies by adding the following sentence after point 3.5:

“Non-conventional methodologies, genetic tools and HTS methodologies, are less labour demanding, improve the speed and capacity of testing, and discovering of novel bioactive molecules. They potentially allow the decrease of the needed time for obtainment of new pharmaceuticals, which is a fundamental aspect in the lengthy process of drug discovery.”

While discussing about dereplication strategy under section 4, more compounds should be reviewed in terms of application of these strategies. Similarly, compounds isolated using genomic and metabolomic mining strategy should be reviewed in more details rather than just explaining the databases.

R: Examples of natural products discovered through genomic and metabolomic mining are provided in the manuscript and, as suggested, this strategies further developed. As such, the following sentence was added in sections 3.4 and 4.3:

This approach is already successfully assisting in discovering and characterize new natural products, as is the case for the macrolactams lobosamide A, B and C [72], which were characterized with the help of antiSMASH and the polythioamide neothioviridamide [85], which biosynthetic cluster belongs to a cryptic RiPP and was heterogeneously expressed in S. avermitilis (discussed in sections 2.2.1 and 2.2.3).

“GNPS molecular networking has been successfully applied to the discovery of several bioactive natural products, including the antimicrobial angucycline-derived polycyclic aromatic polyketide lugdunomycin (65) [170], and the aromatic polyketide accramycin A (66) [171], both isolated from Streptomyces sp., as well as the antibacterial chlorinated cyclic hexapeptides noursamycins A-F (67-72), obtained from cultures of Streptomyces noursei NTR-SR4 [172].”

“The SMART technology in combination with GNPS was successfully applied recently to the discovery of symplocolide A (73), a new chimeric swinholide-like macrolide with cytotoxic properties against the NCI-H460 human lung cancer cell line obtained from extracts of the filamentous marine cyanobacterium Symploca sp. [173].”

Kindly reorganize the structures in the continuous number format rather than alphabets separate for each figure.

R: The structures were accordingly numbered.

The introduction paragraph is very long and need to be reorganized. It lacks the logical flow.

R: The introduction paragraph was sectioned to allow an easier reading and it has also been organized to allow a better flow. We hope to have complied with the reviewers suggestion.

Timeliness is an essential factor for review articles. I would like to see the timeframe this review article covers.

R: We are grateful for the suggestion and have included the following sentence in the abstract and:

“This review highlights information that has been published since 2014, showing the current relevance of marine bacteria for the discovery of novel natural products.”

Reviewer 3 Report

The current manuscript review the bioactive secondary from marine bacteria. Overall, the manuscript prepared well and neat. However, some points require the attention of the authors as below;

1) The Abstract seems as an introduction rather than an abstract. This part should be re-written.

2) Many previous reviews cover a similar topic. The difference should be suggested clearly in an introduction. The current introduction is weak for the point.

3) Referring compound with a letter is not common in the chemistry field. In most cases, sequential numbers in bold are used.

4) In figure legends, strain names are not in Italic.

5) Please check the References. Some have hyperlink, and some doesn't have the date of access.

Author Response

We thank the reviewer for pointing certain weaknesses in the manuscript. Bellow (in bold) are our answers for the points raised:

The current manuscript review the bioactive secondary from marine bacteria. Overall, the manuscript prepared well and neat. However, some points require the attention of the authors as below;

1) The Abstract seems as an introduction rather than an abstract. This part should be re-written.

R: The abstract was re-written to comply with the suggestion. By reducing introductory aspects and putting emphasis on what was discussed in the manuscript, we hope to please the reviewer.

2) Many previous reviews cover a similar topic. The difference should be suggested clearly in an introduction. The current introduction is weak for the point.

R: This review is quite different from others because it puts together the health risks faced presently with the whole drug discovery process, making a revision of literature since 2014. The following sentence was added to the manuscript at the end of the introduction

“After pointing out the fundamental health risks faced by humankind, this review explores the contribution of marine bacterial derived natural products in the fight against cancer, antimicrobial resistance and viruses, with an update highlighting information from the literature published since 2014. Moreover, the technicalities needed for discoveries will be detailed.”

3) Referring compound with a letter is not common in the chemistry field. In most cases, sequential numbers in bold are used.

R: Done

4) In figure legends, strain names are not in Italic.

R: Done

5) Please check the References. Some have hyperlink, and some doesn't have the date of access.

R: Done

Reviewer 4 Report

The manuscript discusses the various health problems faced by humankind, including viral infectious outbreaks, a drastic rise in antibiotic resistance and cancer incidence, pointing out that the ocean is an important way to solve the public health crises. By analyzing the research status of marine bacterial metabolites, it is pointed out that marine bacteria can produce antimicrobial, anticancer, antiviral and other biologically active natural products. It further discusses the research methods for discovering active marine bacteria and novel compounds to avoid the re-separation of known compounds. In general, the manuscript will draw researchers' attention to marine bacteria to a certain extent through the analysis of the current status of marine bacterial metabolite research.

1. The natural products part analyzes the current research situation of natural products of antibacterial, antiviral and anticancer marine bacteria, using a variety of active compounds as examples, but the order is not clear enough. It is better to summarize and classify.

2. For the natural products section, specific cytotoxic activity data can be given, which is more convincing.

3. The structure of Figure 3 overlaps with the name. The position of the structure in Figures can be adjusted appropriately to make it clearer.

4. There are some writing details in the text, and the format is not uniform. For example, Neoabyssomicin D in Table 2 should not be bolded, the space in the brackets (Fig. 5K-N) and (Fig. 5 O) etc.

5. The conclusion is a little bit tedious. The theme can be summarized briefly, and the opinions of authors can be put forward.

Author Response

We thank the reviewer for pointing out weaknesses in the manuscript. Bellow follows (in bold), our answers to the points raised by the reviewer.

The manuscript discusses the various health problems faced by humankind, including viral infectious outbreaks, a drastic rise in antibiotic resistance and cancer incidence, pointing out that the ocean is an important way to solve the public health crises. By analyzing the research status of marine bacterial metabolites, it is pointed out that marine bacteria can produce antimicrobial, anticancer, antiviral and other biologically active natural products. It further discusses the research methods for discovering active marine bacteria and novel compounds to avoid the re-separation of known compounds. In general, the manuscript will draw researchers' attention to marine bacteria to a certain extent through the analysis of the current status of marine bacterial metabolite research.

  1. The natural products part analyzes the current research situation of natural products of antibacterial, antiviral and anticancer marine bacteria, using a variety of active compounds as examples, but the order is not clear enough. It is better to summarize and classify.

R: In general, the principle for the order in which the molecules appear in the text is based on phylogeny of the bacteria. I hope that now it is clear for the reviewer the logic behind our sequence. In the table 3 we have corrected the order of the molecules relative to the bacteria phylogeny.

  1. For the natural products section, specific cytotoxic activity data can be given, which is more convincing.

R: We do not completely understand the point raised by the reviewer because, for many of the compounds, cytotoxicity is provided. If the reviewer asking specifically to the level of activity of the molecules (i.e. IC50 values), we had problems in providing this data, because of the discrepancies in the different reported works. However, and when possible, we have referred the degree (high, medium, low) level of cytotoxicity for each molecule.

  1. The structure of Figure 3 overlaps with the name. The position of the structure in Figures can be adjusted appropriately to make it clearer.

R: Done

  1. There are some writing details in the text, and the format is not uniform. For example, Neoabyssomicin D in Table 2 should not be bolded, the space in the brackets (Fig. 5K-N) and (Fig. 5 O) etc.

R: Done

  1. The conclusion is a little bit tedious. The theme can be summarized briefly, and the opinions of authors can be put forward.

R: Thank you for pointing this aspect. We agree with the reviewer and revised the conclusions. The new text is as follows:

“It is evident that great heath challenges are still in need of novel pharmaceutical responses. Resistance of bacteria and fungi to antibiotics, of parasites like malaria to the available treatments, unavailability of medicines to treat new viruses and efficient drugs for cancer treatment, require additional research and development focus in discovering novel bioactive molecules. Oceans, home to a substantial portion of the world’s biodiversity, are still underexplored, and are an important source of drugs and drug leads. Since 2014, several novel marine bacterial natural products, obtained mainly from Actinobacteria, Firmicutes and Proteobacteria, were described and reported in this review. In supplement to conventional methods, high-throughput analysis, like genome and metagenomic analysis and HTS, and the use of new dereplication and identification tools, will foster the discovery of new leads. This review shows that marine bacteria are key to the development of new pharmaceuticals, especially if combined with a rational, high-throughput approach.”

Round 2

Reviewer 2 Report

The review article titled "From ocean to medicine: some pharmaceutical applications of metabolites from marine bacteria" by Jose et al., has improved substantially after revision to be considered for publication.